# OpenReview forum: "EnIGMA: Interactive Tools Substantially Assist LM Agents in Finding Security Vulnerabilities"
_ICML.cc/2025/Conference — ICML 2025 poster_

### Official Review · Reviewer_FxPC · 2025-03-07

**Overall Recommendation:** 3

**Summary:**

This paper presents EnIGMA, an LM agent enhanced with Interactive Agent Tools (IATs) to solve CTF challenges, achieving state-of-the-art results. Their experiments use 390 challenges from diverse benchmarks to evaluate EnIGMA with different LLMs. They also provided several ablation studies and analyses to demonstrate the effectiveness of EnIGMA and model behaviors.

**Claims And Evidence:**

Partially.

I wonder about their claim about SOTA performance.
In Section 3.2 and Table 2, they include the previous best methods without a clear demonstration of the comparison. I wonder what the particular method used in previous best methods and if they used the same LLM as the agent? Does the superiority come from the better LLMs or the proposed framework?

**Essential References Not Discussed:**

None.

**Experimental Designs Or Analyses:**

Yes.
I found some experimental results that need more analysis and discussion.
1. Table 1: The LM summarizer improves performance by 2.6% over the simple summarizer, but removing both summarizers only reduces performance by 1.3%. This paradox is unexplained.
2. Table 1 shows that ablating IATs reduces overall performance by 2.1%, but in the web category, performance improves by 3.45% (Table 10). This suggests the current IATs may be suboptimal for web challenges.
3. For their criteria for solution leakage, I do not agree with the second condition. The LLM may have learned a large number of flags and found the solution by method of exclusion.
4. Line 431, the experiment with unseen changes is unclear and doesn't cite the data source appropriately. I wonder what the tasks are and what the differences are between the unseen task and existing benchmarks.

**Methods And Evaluation Criteria:**

Yes.
1. I would like to ask for more clarification of tool design: In Section 2.1, specify why only gdb and pwntools were chosen for IATs. Are tools like tshark or nikto (mentioned in Appendix D and Figure 10) accessible to the agent?
2. It is not clear to me how the cost metric is calculated. Did you count the API cost for GPT models? How about LLaMA?

**Other Comments Or Suggestions:**

Please see the comments in the above sections.

**Other Strengths And Weaknesses:**

Strengths:
1. They evaluate their proposed framework with 390 challenges on four benchmarks (NYU CTF, CyBench, etc.) and diverse LLMs. It shows the generalizability of their framework.
2. The discussion of "soliloquizing" and leakage quantification provides useful insights into LM evaluation pitfalls.

Weaknesses:
Please see the comments in the above sections.

**Questions For Authors:**

Please see the comments in the above sections.

**Relation To Broader Scientific Literature:**

Lack of novelty. The tool use of agent and conversation summarization have been explored in previous works. I didn't find new methodology contributions, although I acknowledge the engineering contributions of this paper.
As the main contributions are the engineering aspect, I expect that the paper has more discussion about the code release and ethical considerations.

**Theoretical Claims:**

The paper doesn't have theoretical claims.

---

> ### Author Rebuttal · Authors · 2025-03-31
>
> Thank you so much for your time and consideration. You’ve brought up excellent points in your feedback that we address below.
>
> **Q1: Previous best methods comparison - does the superiority come from the better LLMs or the proposed framework?**
>
> To address your concern, we present agent performance across different benchmarks while using the same LMs: On the NYU CTF benchmark with GPT-4 Turbo, EnIGMA achieves a 7% solve rate vs. NYU Agent’s 4%. On CyBench with Claude 3.5 Sonnet, EnIGMA reaches 20%, outperforming CyBench’s 17.5%. With Llama 3.1 405B Instruct, EnIGMA scores 10% vs. CyBench’s 7.5%. All agents use a ReAct framework with access to a Linux terminal in a Dockerized environment. CyBench, specifically, runs on Kali Linux, where the agent can benefit from a lot of pre-installed security tools. These gains, across benchmarks and models, show that EnIGMA’s framework – not just LM choice – drives performance improvements. We will clarify it in Section 3.
>
> **Q2: Why only gdb and pwntools were chosen for IATs? Are tools like tshark or nikto accessible to the agent?**
>
> We selected the most common tools which were unsupported in current LM agents based on our experiments on the development set. Tools like `tshark` and `nikto` remain accessible but are not part of IATs because they already have well-structured CLIs that can be invoked directly as shell commands without requiring interactivity. We will clarify this in the paper.
>
> **Q3: How is the cost metric calculated?**
>
> The cost metric is calculated per solved challenge based on API calls, taking into account input & output tokens and model-specific pricing. OpenAI and Anthropic models use their official pricing, while Llama models follow Together AI's API rates. Details are in Appendix C.2, and we will further clarify the cost calculation in the paper.
>
> **Q4: The LM summarizer improves performance by 2.6% over the simple summarizer, but removing both summarizers only reduces performance by 1.3%.**
>
> The LM summarizer and the simple summarizer are actually two distinct modules. The LM summarizer condenses the previous action’s output into a short summary (so that the agent can process long outputs from tools such as a decompiler). The simple summarizer shows the first 100 lines of the last action’s output. We show the results of the simple summarizer, which degrades the baseline agents performance by 2.6%, just to show that a simple approach to summarizing doesn’t perform well. On the other hand, our LM summary tool improves performance by 1.3% over the baseline. We apologize for the confusion this caused and will remedy this in the next version of our paper by clarifying the role of the simple summarizer as a baseline summarizer, that should not actually be used in practice.
>
> **Q5: IATs may be suboptimal for web challenges.**
>
> This is correct. As noted in Section 4.1 (Line 316), the performance increase in the web category suggests that the current IATs may be less suited for these types of challenges. At the same time, this result highlights the effectiveness of interactive tools in the categories where they are most relevant - crypto, pwn, and rev - where their presence contributes to the agent’s success. There are a large number of ways to further expand IATs to handle a wider variety of tasks, and so we leave that for future work.
>
> **Q6: Solution leakage criteria and finding the solution by method of exclusion.**
>
> Our second condition specifically assesses whether the flag appears in any observations, which are the outputs generated by the environment. In the scenario you described the flag would not appear in any of the environment’s outputs. According to our definition, this case would indeed be classified as solution leakage.
>
> **Q7: Unseen challenges - data source and differences with existing benchmarks.**
>
> Thank you for bringing this to our attention. We will cite the GitHub repository with these new challenges, which were part of the qualification round of the 2024 CSAW competition, following the same competitions as the NYU CTF benchmark but from different years. These challenges span the same six categories: 5 crypto, 4 forensics, 3 web, 4 rev, 4 pwn, and 1 misc. The key difference is their release date – September 2024, after all models’ training cutoffs. EnIGMA with Claude 3.5 Sonnet solves 2 of 21 challenges suggesting that it can extrapolate to new problems that the underlying LM has not encountered during training.
>
> **Q8: Novelty.**
>
> Please see reviewer FRVw, Q5.
>
> **Q9: Code release and ethical considerations.**
>
> We are committed to open-sourcing our code and have included an anonymized repository with all experimental artifacts in the supplementary materials. Given the cybersecurity focus, we address ethical considerations in the Impact Statement (page 9) and have disclosed our findings to model providers to ensure awareness of potential safety implications.
>
> Thank you again, your constructive feedback is valuable in refining our work.

---

> > ### Comment · Reviewer_FxPC · 2025-04-05
> >
> > Thanks for the response.
> > I found many of my concerns have been addressed. I hope the authors can incorporate these into the final version of the paper.
> >
> > For Q1, I hope the author can summarize the SoTA comparison results in a table.

---

> > > ### Author Response · Authors · 2025-04-06
> > >
> > > We thank reviewer FxPC for their response and score increase.
> > >
> > > > I hope the authors can incorporate these into the final version of the paper.
> > > For Q1, I hope the author can summarize the SoTA comparison results in a table.
> > >
> > > We are committed to incorporate these changes into the final version of the paper. We will also include a table summarizing the SoTA comparison between our agent and previous best methods using the same LMs.
> > >
> > >  Thank you for your valuable feedback.

---

### Official Review · Reviewer_NpAu · 2025-03-13

**Overall Recommendation:** 3

**Summary:**

The paper proposes EnIGMA, an LM agent designed for CTF challenges. EnIGMA is built based on SWE-agent for code generation, which is based on the ReAct framework. On top of SWE-agent, EnIGMA incorporates actions and tools specially designed for the CTF challenges, including a debugger and a remote connection server tool. EnIGMA is evaluated on four CTF challenges (3 public and 1 self-created) and demonstrates clearly better performance than previous state-of-the-art.

**Claims And Evidence:**

I didn't find faulty claims.

**Essential References Not Discussed:**

I didn't find important missing references.

**Experimental Designs Or Analyses:**

The experimental design follows the standards for CTF challenges.

**Methods And Evaluation Criteria:**

The proposed method makes sense in general. The evaluation is also rigorous from my point of view.

**Other Comments Or Suggestions:**

The system can implement a memory module to store previously solved cases. The memory entries can be retrieved to further enhance the performance.

**Other Strengths And Weaknesses:**

The paper is generally well-written and the results are convincing.

My major concern is the novelty of the method. Incorporating specially designed tools is a typical design when adapting generic agents to specific tasks. In this work, it is hard to claim that the performance gain is due to the agent workflow design rather than harnessing the power of the tool.

**Questions For Authors:**

If for the previous methods, the same debugger is hardcoded into the system, will they achieve much better results than they did before?

**Relation To Broader Scientific Literature:**

The application of ReAct-based agent to the CTF challenges is reasonable, demonstrating the power of LM agents in solving complex tasks. The results are expected, as LM agents with tools should perform better than naive LLM prompting.

**Theoretical Claims:**

There are no theoretical claims in this paper.

---

> ### Author Rebuttal · Authors · 2025-03-31
>
> We sincerely thank the reviewer for their feedback on our work and for finding our results convincing. We address each of your concerns below.
>
> **Q1: Novelty of the method and whether the performance gain due to the agent workflow design or harnessing the power of the tools.**
>
> In EnIGMA, we are the first to show how to enable an LM agent to utilize ***interactive tools***, such as a debugger and a server connection tool. To facilitate the use of such tools, we developed the IATs framework, which also provides extendability for future research on other interactive tools (Section 2). This approach allows the agent to perform tasks it previously could not, even when tools are directly installed on the environment. As a result, EnIGMA achieves state-of-the-art performance on three out of the four benchmarks we tested, even when using the same LMs and methods as in previous approaches (see reviewer FxPC, Q1). Our comprehensive empirical analysis (Section 4) explores how the LM agent utilizes the framework, where we demonstrate the agent’s effective use of interactive tools. Lastly, we also ***uncover the surprising soliloquizing phenomenon***, which provides valuable insights into the design and evaluation of future LM agents. We hope this clarifies the novelty and impact of our method, and we are grateful for your feedback.
>
> **Q2: The system can implement a memory module to store previously solved cases. The memory entries can be retrieved to further enhance the performance.**
>
> This is a valuable suggestion. Prior research on LM agent frameworks has shown that a memory module can improve performance. We leave this as a direction for future research.
>
> **Q3: If for the previous methods, the same debugger is hardcoded into the system, will they achieve much better results than they did before?**
>
> In the reverse engineering (rev) category – where the debugger is most frequently used (Figure 9) – Table 10 indicates a 3.84% drop in solve rate when the interactive tools (debugger and server connection) are removed. We know that the debugger was invoked in 8.1% of these tasks, and the server connection was used just in 3.3% of cases, we can therefore summarize that removing only the debugger would harm performance.
> In addition, Table 1 shows that removing both the debugger and server connection tools reduces the overall solve rate by 2.1% across all categories in all four benchmarks.
> In the initial paper we didn’t have an ablation study that just ablates the debugger away, but we agree that this is an important number to have and will run these experiments for the final version of the paper.
>
> Thank you once again for your valuable feedback.

---

### Official Review · Reviewer_t5tU · 2025-03-14

**Overall Recommendation:** 4

**Summary:**

The paper describes a new and improved agent for solving computer security Capture the Flag challenges.

**Claims And Evidence:**

Mostly.  There is one claim about interactive tools that I think is overstated (see Other comments below).  The evaluation of leakage has some limitations.

**Essential References Not Discussed:**

Not that I know of.

**Experimental Designs Or Analyses:**

The experimental designs seem reasonable.  I would have been interested if there was a more convincing way to evaluate leakage, but that's challenging to do, so it's not a surprise that it is difficult to get a solid grip on it.

**Methods And Evaluation Criteria:**

Yes.  I think the benchmarks and evaluations are reasonable and appropriate and support the paper's goals.

**Other Comments Or Suggestions:**

I don't understand what "main shell" means.  Does this mean a connection to a Unix shell (e.g., bash)?  Or does this mean an agent running a REACT loop?

Fig 2: I don't understand how this demonstrates a session running in parallel with the main shell.  What part of Fig. 2 is the main shell and what part is the separate session?  What does "bash-\\$" mean?  Does that prompt mean that the FTP connection has terminated and now the session involves the agent interacting with bash?  Or is the FTP connection still open and the agent is still interacting with it?  If so, why provide input like "bash-\\$" that normally indicates back to the command-line shell rather than in a program like a FTP client?

Sec 2.2: What is the input to the summarizer?  Does the summarizer receive the prior thought and action (e.g., "Let's start by decompiling...")?  Does it receive the initial context about the problem (e.g., the text of the problem statement for this CTF problem)?  Or does it only receive the tool output and nothing else?

Sec 2.3: Please provide more detail on this.  How many demonstrations per step?  How were they selected?  Can you report the average number of guidelines and a histogram on them, and show some randomly selected examples of guidelines?

Table 1: On what dataset is this measured?

Table 2: I recommend showing Table 2 earlier, and moving Table 1 later.  First, show the main overall results on effectiveness of your method.  Save the ablations for later.

Sec 4.1: Could we achieve a higher pass rate, if we halt the process after 20 steps (if it hasn't succeeded yet), reset everything, and restart anew?  In other words, if in one run, the agent never succeeds, might there be another luckier trajectory that does succeed, and if we try multiple trajectories, does it increase the probability that at least one succeeds?  It might be better to try 5 times, each for only steps, than to try once, for 100 steps.

Sec 4.1: I don't agree with the conclusions here.  The quantitative results don't support the claim that "Proper interactive interfaces are crucial".  The performance drop with IATs is only 2 percentage points (about 10%).  That's not "crucial".

Sec 4.2: The criteria used to measure leakage seems a bit too narrow to me.  I can imagine that if the challenge was in the training set, the LLM might not have memorized the flag but might have memorized the solution approach.  (For instance, maybe the LLM training data included a writeup from one of the contest participants on their blog.)  I think it would be helpful if you had a way to measure leakage in a more convincing way.

Sec 4.2, unseen challenges: It might help to mention which of the 4 datasets has the most similar distribution to the set of 21 unseen challenges, so we know what to compare to.

Typos: "miscallaneous", "challenges.."

**Other Strengths And Weaknesses:**

This work is helpful, because it helps us understand the risks of LLM agents and the balance of power between attackers vs defenders (do stronger agents help attackers more, or help defenders more?).

This work might also be useful in the future for system defenders, because solving CTF puzzles is related to finding vulnerabilities.  These techniques might improve effectiveness of agents at finding vulnerabilities, and system defenders could use those methods to find and then patch vulnerabilities in their systems.

The paper is well-written and easy to follow.

**Questions For Authors:**

I have no particular prioritization

**Relation To Broader Scientific Literature:**

This continues a line of work exploring using LLM agents to solve CTF puzzles, and improves upon past work.  It finds that better design of tool bindings improves overall performance.

**Theoretical Claims:**

No theoretical claims made.

---

> ### Author Rebuttal · Authors · 2025-03-31
>
> Thanks for finding our work helpful. Your thorough review of the paper and suggestions has been helpful to clarify several details.
>
> **Q1: What "main shell" means. Does this mean a connection to a Unix shell (e.g., bash)? Or does this mean an agent running a REACT loop?**
>
> “Main shell” is a connection to a unix shell, specifically in our case it is a bash shell. We will clarify this in the revised paper.
>
> **Q2: Fig 2: How does this demonstrate a session running in parallel with the main shell?**
>
> In our setup, the agent always has access to the main shell, which is represented as `bash $-`. The interactive tool runs as a separate process in the environment and is displayed to the agent as a distinct line, labeled `(Interactive session …)`. The agent can interact with this parallel process through special interfaces, which are accessible from the main shell. These interfaces are described in detail in Table 7. In Figure 2, the agent demonstrates accessing the parallel process by using the `connect_sendline` interface to send a command to the FTP server. Then, the agent uses the `decompile` within the main shell to decompile a binary, while the FTP connection is still maintained. We will update the caption of Figure 2 to better reflect this interaction.
>
> **Q3: Sec 2.2: What is the input to the summarizer?**
>
> As outlined in Section 2.2, the input to the simple summarizer consists of the long observation, which is saved to a file and then opened using SWE-agent's file viewing interface. This allows the agent to view the first 100 lines, and the agent can then scroll or search through the file as needed. For the LM summarizer, the input includes the challenge context – the challenge's name, category, and the CTF problem text – as well as the last action performed by the main agent and its resulting output. More detailed information about all the prompts can be found in Appendix G, with the LM summarizer prompts specifically described in Appendix G.2.
>
> **Q4: Sec 2.3: Demonstrations and guidelines details**
>
> Each challenge category (crypto, rev, misc, web, forensics, pwn) has its own demonstration, where the crypto category has two demonstrations, while the other categories have one demonstration each. These demonstrations were selected randomly from the failed challenges in the development set. Specifically, we ran our agent without any demonstrations on the challenges in the development set, then we randomly selected failed challenges to include in the demonstrations and manually created successful trajectories to include as demonstrations. The guidelines, which were derived through trial and error from runs on the development set, are based on manual observations from failed attempts. We have a total of 9 general guidelines, along with 5 additional guidelines specifically for the debugger. All of these guidelines are provided in Appendix G.1, Figure 12 (Line 1516).
>
>
> **Q5: Table 1: On what dataset is this measured?**
>
> The results are aggregated results on all four benchmarks - NYU CTF, CyBench, InterCode-CTF and HTB.
>
> **Q6: Table 2: I recommend showing Table 2 earlier, and moving Table 1 later.**
>
> Thank you for pointing this out - we will change this in the revised paper for a more streamlined reading.
>
> **Q7: Sec 4.1: Could we achieve a higher pass rate, if we halt the process after 20 steps (if it hasn't succeeded yet), reset everything, and restart a new?**
>
> As our pass@1 results suggest, our agent succeeds fast and fails slow (Figure 4). The proposed suggestion to restart the trajectory after X steps, given our results, may be helpful as the agent is unlikely to succeed when it reaches an impasse in one trajectory. We leave this as a future research direction.
>
> **Q8: Sec 4.1: The quantitative results don't support the claim that "Proper interactive interfaces are crucial".**
>
> Thanks for the observation, we will change this to “Proper interactive interfaces enhance performance”.
>
> **Q9: Sec 4.2: The criteria used to measure leakage, and whether the agent memorizes a solution approach from training data rather than the flag.**
>
> The leakage issues are something that we address extensively in Section 4.2, using solution leakage quantification, uncovering soliloquizing phenomena which relates to solution approach leakage and by measuring our agent using new challenges released after training cutoff date of all of the models used in our evaluations. Please refer to reviewer FRVw, Q1.
>
> **Q10: Sec 4.2, unseen challenges: It might help to mention which of the 4 datasets has the most similar distribution to the set of 21 unseen challenges, so we know what to compare to.**
>
> Thanks for observing the missing information, please refer to reviewer FxPC, Q7.
>
> **Q11: Typos: "miscallaneous", "challenges.."**
>
> Thank you for catching the typos, we will fix these in the next version of the paper.
>
> We hope to have addressed the concerns raised and will update the paper accordingly. We appreciate your thoughtful review.

---

> > ### Comment · Reviewer_t5tU · 2025-04-07
> >
> > Thank you for your response about data leakage.  I had indeed missed that experiment, and I think it is responsive and helpful.  I appreciate how you have dealt with data leakage; ruling out data leakage is very challenging, but I think the analysis here has done a good job of addressing the concern, or as good as is reasonably possible given the challenges in this area.  I also appreciate the addition of detailed comparison to state-of-the-art schemes.  I think that will further strengthen the paper.  I continue to recommend accepting this paper.

---

> > > ### Author Response · Authors · 2025-04-08
> > >
> > > We thank reviewer t5tU for their response and for finding the data leakage analysis and experiments helpful.
> > >
> > > We will incorporate the valuable feedback raised during the rebuttal in the final version of the paper.

---

### Official Review · Reviewer_FRVw · 2025-03-19

**Overall Recommendation:** 3

**Summary:**

This paper presents EnIGMA, an LM agent designed for autonomously solving Capture The Flag (CTF) challenges.
- The authors introduce Interactive Agent Tools (IATs), which enable the LM agent to execute interactive cybersecurity tools such as debuggers and remote server connection utilities. These tools address key limitations in prior LM-based cybersecurity agents, which lacked the ability to use interactive command-line utilities.
- The authors evaluate EnIGMA on 390 CTF challenges across four benchmarks (NYU CTF, InterCode-CTF, CyBench, and a collected HackTheBox (HTB) dataset), reporting state-of-the-art performance on three of these benchmarks.
- Additionally, they introduce a method for quantifying data leakage and identify a novel phenomenon termed soliloquizing, where the LM hallucinates entire challenge solutions without environmental interactio

**Claims And Evidence:**

- EnIGMA achieves state-of-the-art performance on multiple CTF benchmarks.
  - The paper provides empirical results comparing EnIGMA to prior LM-based agents, showing significant improvements in solved challenges. However, it is unclear whether the performance gain is due to genuine advancements in reasoning or data leakage from training corpora.

**Essential References Not Discussed:**

no

**Experimental Designs Or Analyses:**

strengths:
- Ablation studies show that IATs, summarizers, and demonstrations contribute to performance improvements.
- Diverse benchmarks ensure broad evaluation across different cybersecurity challenges.


Weaknesses:
- Unclear statistical significance: The paper lacks confidence intervals or statistical tests to validate improvements.
- Data leakage analysis is incomplete: The authors attempt to quantify leakage but cannot verify whether training data contamination influenced results.

**Methods And Evaluation Criteria:**

The paper adopts CTF benchmarks to evaluate cybersecurity-focused LM agents, which is a reasonable choice for testing the agent’s practical problem-solving ability. However, the evaluation lacks sufficient details on dataset splits, agent hyperparameters, and exact experimental conditions.

**Other Comments Or Suggestions:**

see weakness

**Other Strengths And Weaknesses:**

strengths:
The integration of IATs for debugging and server interaction is a meaningful addition to LM agent capabilities.

weakness:
- The paper is difficult to follow, with unclear explanations of contributions and inconsistent structuring.
- The work is mostly engineering-driven rather than presenting new conceptual frameworks.

**Questions For Authors:**

see weakness

**Relation To Broader Scientific Literature:**

The paper extends work on LM agents for cybersecurity by introducing interactive tools.

**Theoretical Claims:**

The paper does not introduce new theoretical foundations but instead focuses on empirical improvements through tool integration.

---

> ### Author Rebuttal · Authors · 2025-03-31
>
> Thank you for your interest in our research and the acknowledgment of IATs as a meaningful addition to LM agents. We greatly appreciate your feedback and insights which will help us improve our work. We’ve addressed your concerns below:
>
> **Q1: Data leakage analysis is incomplete + It is unclear whether the performance gain is due to genuine advancements … or data leakage from training corpora.**
>
> Addressing whether agents solve problems through reasoning or by relying on memorization remains a challenge in LM evaluations. Yet, we tackle this issue extensively (Section 4.2, Table 3 and Appendix E) by quantifying solution leakage where the flag is submitted without appearing in prior observations and by uncovering the soliloquizing phenomenon (hallucinated observations), which relates to data leakage of the solution approach (Figure 7).
>
> Moreover, results on challenges released after the training cutoff of all models used in our evaluations (Section 4.2, Line 431) shows that EnIGMA is able to solve 2 out of 21 challenges that it *could not* have seen before. Combined with comparison between EnIGMA to other agents using the same LM versions (see reviewer FxPC, Q1) we can attribute EnIGMA’s performance gain to the novel agent improvements that we introduced in this work and not to solution leakage.
>
> **Q2: The evaluation lacks sufficient details on dataset splits, agent hyperparameters, and exact experimental conditions.**
>
> Our evaluation involves four test benchmarks: NYU CTF, CyBench, InterCode-CTF, and HTB, as well as a self-created development benchmark. We discuss each of these benchmarks in Section 3, and we provide further details on the benchmarks and experimental conditions in Appendices B and C, including the LM parameters (model versions, temperature, and nucleus sampling values). For the LM agent configuration, we outline the interfaces and environment used during evaluations in Appendix D, and we also provide all the prompts used in the evaluation in Appendix G. We adopted the default parameters from SWE-agent. Demonstrations were provided per challenge category: two for the crypto category, and one for each of the other categories. We appreciate your comment on this and will ensure this information is presented more clearly in the next version of our paper.
>
> **Q3: Unclear statistical significance.**
>
> We fully agree that measuring statistical significance is important. As you mentioned in the strengths, our ablation studies (Table 1) along with the analysis in Section 4.1 demonstrate that the performance improvements can be attributed to the IAT framework we introduced. Specifically, using these interactive tools, the agent solves challenges in an average of 11.5 turns, which is 22.8% faster than the 14.9 turns required when they are not used (p-value: 0.019). Combined with the results shown in Figure 4, which highlight that the agent is more likely to succeed quickly and fail slowly, we can claim the performance gain to the proposed interactive agent tools framework. We will incorporate this statistical analysis in the revised paper.
>
> **Q5: The work is mostly engineering-driven rather than presenting new conceptual frameworks.**
>
> Our primary contribution is the introduction of a novel agent for the cybersecurity domain, designed with specialized tools and interfaces that enhance its ability to solve CTF challenges.
>
> We are the first to demonstrate how an agent can utilize ***interactive tools***, such as a debugger and a server connection tool. We developed the IATs framework to facilitate future research in enabling LM agents to use such tools (Section 2.1). These new tools allow EnIGMA to achieve state-of-the-art performance across three of the four benchmarks.
>
> Lastly, we provide a comprehensive empirical analysis of the agent's behavior (Section 4), where we ***uncover the unexpected soliloquizing phenomenon***. This finding provides valuable insights that can inform the design and evaluation of future LM agents. We hope this clarifies the contributions of our work.
>
> **Q6: The paper is difficult to follow, with unclear explanations of contributions and inconsistent structuring.**
>
> We apologize for any confusion caused by the structure of the paper. Our contributions are outlined in both the introduction and conclusion, and as an answer to Q5 above. These contributions are elaborated in the relevant sections of the paper: Section 2 introduces our agent, including the IATs and summarizers; Section 3 details the development set and experimental setup; and Section 4 presents the empirical analysis including solution leakage and soliloquizing phenomenon. We are not sure what you mean by inconsistent structuring, and would be happy to remedy this if you briefly explain the concern.
>
> We appreciate your insights once again and are committed to improving the clarity of the paper.

---

> > ### Comment · Reviewer_FRVw · 2025-04-09
> >
> > Thanks for the response. I found many of my concerns have been addressed. I will update my final score!

---

### Decision · Program_Chairs · 2025-05-01

**Decision:**

Accept (poster)

**Comment:**

The paper proposes a new agent design for solving CTF security challenges.
Reviewers agreed that the paper does a good job at motivating its design, and improves the state-of-the-art in this domain.
The approach to mitigating data leakage was also well received.
Overall, I recommend acceptance for this paper.